# SELF-CHOOSE: LEVERAGING DIVERSE REASONING SOLUTIONS TO SELF-CORRECT MULTIMODAL LARGE LANGUAGE MODELS

## ABSTRACT

In the past few years, Multimodal Large Language Models (MLLMs) have achieved remarkable advancements in reasoning while still suffering from mistakes. Some existing approaches on LLMs self-correct the answers without external feedback, proven limited in reasoning. We revisit these previous approaches and propose an improved effective strategy dubbed **Self-Choose** to teach MLLMs to utilize diverse reasoning solutions to self-correct reasoning. Our approach first employs various reasoning methods to generate candidate answers. Then, it evaluates them by comparing the reasoning processes and candidate answers to choose the optimal solution. Finally, it outputs the best candidate or reflects to generate an improved solution if all the answers are deemed inaccurate. We evaluate our method on multiple datasets with mainstream foundation models including LLaVA and Gemini. The extensive experiments show that Self-Choose achieves consistent improvements on different benchmarks and metrics. We hope this study will promote future research on self-correction and its application across various tasks.

## 1 INTRODUCTION

In the past few years, Multimodal Large Language Models (MLLMs) have experienced unprecedented development (Alayrac et al., 2022; Dai et al., 2023; Li et al., 2023a; Liu et al., 2023b; Zhu et al., 2024; Reid et al., 2024; Chen et al., 2023). The great success has motivated researchers to explore and promote the reasoning ability of MLLMs (Wang et al., 2024b; Fei et al., 2024). However, MLLMs often suffer from mistakes in reasoning, hiding their wider applications. Although researchers have made some progress in dealing with hallucinations (Yin et al., 2023; Gunjal et al., 2024; Li et al., 2023b; Liu et al., 2024), these methods mainly focus on simple perception problems, *e.g.*, the existence, specific quantity, position and other attributes of objects. It is rarely explored how to effectively correct errors in complex vision reasoning problems, such as vision-question answering involving advanced knowledge, analyzing images with weird content, and solving mathematical problems illustrated with diagrams.

There have been many works on reasoning correction in Large Language Models (LLM) (Madaan et al., 2023; An et al., 2023; Liu et al., 2023c; Gou et al., 2023; Welleck et al., 2023). Self-correction is an area of research among them that has gained widespread attention (Huang et al., 2024). It aims to only use the same model to correct answers without training or assistance from other tools. Previous self-correction methods are mainly based on a three-step strategy, which first generates an initial response, then evaluates it to produce feedback, and refines the response according to feedback. However, several studies show that LLMs struggle to self-correct reasoning (Huang et al., 2024; Stechly et al., 2023; Valmeekam et al., 2023). In this work, we focus on extending these approaches to MLLMs and explore self-correction methods for them.

We conduct experiments on MLLMs with the self-correction method, **Self-Refine** (Madaan et al., 2023; Kim et al., 2023), but it fails to correct vision reasoning. We analyze the results and find that the model sometimes cannot properly identify problems and changes the right answer to a wrong one. To deal with this problem, we come up with a method named **Self-Review**. Self-Review first uses the model to judge whether the answer is right or wrong, then maintains the original answer if judged as right or uses Self-Refine to correct the original answer if judged as wrong. The performance

of Self-Review is better than Self-Refine. However, it still fails to correct answers properly. The experiment results indicate that it is because the model cannot accurately assess the correctness of the answer. A plausible explanation is that such prompting strategies do not provide additional useful information, making it difficult for the model to accurately assess correctness and identify issues solely based on its intrinsic capabilities (Huang et al., 2024).

This is analogous to the "mental set" phenomenon (Jersild, 1927), a widely studied psychological phenomenon. It refers to the cognitive tendency to approach problems in a particular way based on past experiences, learned behaviors, or established habits, which hinders the ability to explore diverse approaches to find the most suitable method to solve the problem (Öllinger et al., 2008; DeCaro, 2016). Similarly, the model fails to correct itself with fixed thinking pattern.

Inspired by the above analysis, we propose an effective strategy termed **Self-Choose** to teach MLLMs to explore diverse reasoning solutions to choose the optimal one. First, the MLLM uses different reasoning methods to solve the problem and get different solutions. The distinct reasoning methods focus on different aspects such as image understanding and text understanding, which can provide different perspectives of insights and serve as additional useful information created by the model itself. Then, the MLLM reviews the different solving processes of these solutions for comparison and reflection to choose the best one, which can help judge the correctness and identify problems. Finally, the MLLM outputs the best solution if it exists. Otherwise, the MLLM will generate a more promising answer according to these inexact solutions.

We evaluate Self-Choose on three vision reasoning benchmarks that span diverse domains: ScienceQA (Lu et al., 2022) for multiple-choice answering, Whoops (Bitton-Guetta et al., 2023) and MM-Vet (Yu et al., 2023) for complicated vision-question answering. Extensive experiments show that our method can effectively improve the reasoning answers of MLLMs such as LLaVA (Liu et al., 2023a) and Gemini-vision (Reid et al., 2024), while other methods can not. Our method is an effective prompting strategy to teach MLLMs to self-correct, which is plug-and-play and can be applied to black-box MLLMs. In addition, our method does not need the assistance of other models or tools, completely relying on the MLLM itself. It shows the potential capability of MLLMs to self-correct.

## 2 RELATED WORKS

**Reasoning methods in MLLMs.** Chain-of-Thought prompting (Wei et al., 2022; Kojima et al., 2022) is a widely used reasoning method, which solves the problem step by step. Several works explore the efficacy of employing Chain-of-Thought on MLLMs (Lu et al., 2022; Zhang et al., 2023b; Wang et al., 2024a). Based on Chain-of-Thought, some multimodal reasoning methods are proposed, which can be categorized into two types. The first type emphasizes image understanding (Mitra et al., 2023; Zhang et al., 2024; Zhou et al., 2024; Gao et al., 2024b), while the other type focuses on text understanding (Zheng et al., 2023). In spite of these reasoning methods, MLLMs still suffer from mistakes when reasoning. Our work leverages comparison between different methods to facilitate self-correcting these errors.

**Correcting reasoning in LLMs.** There are many different ways to correct reasoning in LLMs. Some researchers train or fine-tune the model with the collected high-quality data (Huang et al., 2023; Liu et al., 2023c; An et al., 2023). Some train a corrector to help correct reasoning (Welleck et al., 2023). While others correct reasoning without training with the assistance of other models or tools (Zhang et al., 2023a; Pan et al., 2023; Peng et al., 2023). Different from them, some works use the same LLM to self-correct completely relying on itself. Self-Refine (Madaan et al., 2023) uses the same model to provide feedback for its output and uses it to refine the output, iteratively. However, it performs poorly on reasoning tasks. Several studies indicate that LLMs struggle with self-correcting reasoning (Huang et al., 2024; Stechly et al., 2023; Valmeekam et al., 2023; Liang et al., 2023). However, self-correcting reasoning on MLLMs is less explored. We conduct experiments applying self-correction techniques originally designed for LLMs to MLLMs, only to discover that such techniques fail to facilitate self-correction in reasoning for MLLMs. To deal with it, we propose an effective approach to teach MLLMs to self-correct like humans.

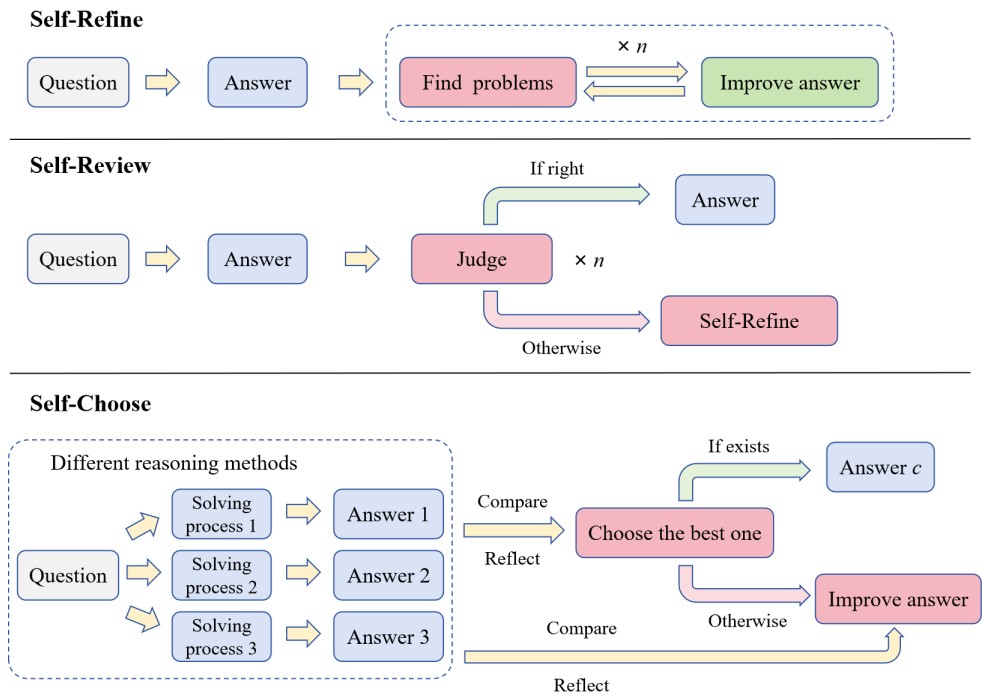

Figure 1: Pipelines of different methods to self-correct: Self-Refine, Self-Review, and Self-Choose. Self-Refine finds problems and refines answers according to the feedback iteratively. Self-Review first judges the correctness of the answer, then keeps the original answer if judged as right or improves the answer with Self-Refine. Self-Choose compares and reflects on the solving processes and answers of different reasoning methods. It chooses the best answer if it exists, otherwise generates a more promising answer according to solving processes and answers of different reasoning methods.

## 3 PRELIMINARIES

### 3.1 SELF-REFINE

**Self-Refine** (Madaan et al., 2023) is a widely used method of self-correction, which uses the same LLM to provide feedback for its output and uses it to refine itself, iteratively. The strategy of Self-Refine consists of three steps: 1. prompt the model to perform an initial answer, which also serves as the result for standard prompting; 2. prompt the model to find problems of its previous answer and produce feedback; 3. prompt the model to answer the original question again with the feedback to get the improved answer.

Although Self-Refine improves performance on diverse tasks, such as sentiment reversal, dialogue response, code readability, and so on, it struggles to self-correct reasoning. Several works have shown that LLMs cannot self-correct reasoning in the way of Self-Refine (Huang et al., 2024; Stechly et al., 2023; Valmeekam et al., 2023). We extend their experiments on MLLMs to explore the ability of MLLMs to self-correct. However, we similarly observe a decrease in performance after Self-Refine. The experiment details can be found in Section 5.4. We analyze the experiment results of Self-Refine, and find that step 2 may mislead the model to nitpick in originally correct answers and fail to identify the real problem. This misguides the model to revise the right answer into a wrong one according to the problem found in step 2.

### 3.2 SELF-REVIEW

To deal with the problems of Self-Refine, we come up with another strategy named **Self-Review**. The strategy is three-step: 1. prompt the model to perform an initial answer; 2. prompt the model to review the initial answer and determine whether it is right or wrong; 3. if the model judges the initial

answer as right, keep the original answer. Otherwise, use steps 2 and 3 in Self-Refine to correct the initial answer. Step 2 in Self-Review is executed multiple times to take a majority vote. Although Self-Review achieves overall better results than Self-Refine, it still cannot improve original answers effectively. We observe that MLLMs are not able to reliably assess the correctness of answers. Detailed experiment results are shown in Section 5.4. Appendix F and G show some examples of Self-Refine and Self-Review, respectively.

## 4 METHOD

Although Self-Refine and Self-Review do not work, we still believe that there may be an effective way to self-correct reasoning for MLLMs. So why do these prompting strategies fail to self-correct reasoning? A feasible explanation is that the designed prompt strategy may not provide any additional useful information for answering the question, making it difficult for the model to properly self-correct solely based on its inherent fixed thinking pattern. Introducing internal feedback can be regarded as adding an additional prompt, which may bias the model toward generating a response tailored to this combined input. It could potentially divert the model from producing the optimal response to the initial prompt, thereby leading to a degradation in performance (Huang et al., 2024). From the above analysis, it can be inferred that introducing additional useful information may assist the model to self-correct reasoning. A natural idea would be to leverage other tools or human supervision to provide supplementary messages to aid in model correction (Zhang et al., 2023a; Pan et al., 2023; Peng et al., 2023). However, this is not our goal. We aim to design an effective strategy to self-correct reasoning completely relying on the model itself.

This dilemma of self-correction is analogous to the psychological phenomena of "mental set". It refers to the cognitive tendency to approach problems in a particular way based on past experiences, learned behaviors, or established habits. In practice, there are usually many different ways and usually one optimal one to solve a problem. However, the mental set hinders diverse thinking to find the most suitable method to solve it.

We make the following analysis on ScienceQA and observe this theory also applies to MLLMs. Using the same model to answer the question three times, if there is one right answer, it is considered correct. We find that with the fixed reasoning method, only 1558 and 1594 answers are correct for LLaVA and Gemini, respectively. While with three distinct reasoning methods, 1624 and 1755 questions are correctly answered for LLaVA and Gemini. It indicates that there may be an optimal method for a single problem. If the model can identify the best solution among different methods, it may achieve more improvement. Therefore, we design a novel prompting strategy to teach MLLMs to choose the best reasoning solution, named **Self-Choose**.

Given an image and a text question, Self-Choose first uses different reasoning methods to answer the question to get candidate answers and solving processes. These methods focus on different aspects, such as image understanding and text understanding. Then it reflects by comparing the solving processes of candidate answers and finally chooses the best candidate answer. However, there may not be a right answer among the candidate answers. So we add the choice that all candidate answers are incorrect. In this situation, the model will be forced to generate a more promising answer according to these wrong candidate answers and their solving processes. The solving processes and answers obtained from various reasoning methods can provide the model with different perspectives for comparison. This is equivalent to the model creating additional useful information by itself, which can assist the model in better assessing the correctness of the answers and identifying issues. We do not adopt the strategy of Self-Refine when generating the more promising answer, in order to avoid the problems discussed in Section 3 and reduce token costs and complexity. All prompts used in Self-Choose are presented in a zero-shot manner. Self-Choose can serve as a training-free prompting strategy to teach MLLMs to self-correct without any assistance of any other models or tools, which is plug-and-play and applicable to black-box MLLMs. Figure 1 shows the pipelines of the three self-correction strategies, Self-Refine, Self-Review and Self-Choose. Next, we describe our method in more detail.

**Generate solving processes and candidate answers.** Given an input which contains an image $\mathbf{x}_{img}$ and a text question $\mathbf{x}_{txt}$, and a MLLM $\mathcal{M}$, Self-Choose selects $n$ reasoning methods $\{\mathcal{F}_i \mid i = 0, 1, ..., n - 1\}$ to guide the model $\mathcal{M}$ to generate $n$ corresponding solving processes $\{\mathbf{s}_i \mid$

$i = 0, 1, ..., n - 1$.

$$\mathbf{s}_i = \mathcal{M}(\mathbf{x}_{img}; \mathbf{x}_{txt}|\mathcal{F}_i), i = 0, 1, ..., n - 1. \tag{1}$$

where $\mathcal{F}_0$ represents standard prompting method and $\mathbf{s}_0 = None$. According to the solving process $\mathbf{s}_i$, the model $\mathcal{M}$ outputs the candidate answer $\mathbf{y}_i$ corresponding to the reasoning method $\mathcal{F}_i$.

$$\mathbf{y}_i = \mathcal{M}(\mathbf{x}_{img}; \mathbf{x}_{txt}, \mathbf{s}_i|\mathcal{F}_i), i = 0, 1, ..., n - 1. \tag{2}$$

**Reflect and choose the best candidate answer.** Next, Self-Choose uses the same model $\mathcal{M}$ to compare and reflect on the solving processes and candidate answers, and finally choose the best candidate answer. Given a prompt $\mathbf{p}_{cho}$ guiding the model $\mathcal{M}$ to choose the best candidate answer, the model $\mathcal{M}$ compares and analyzes the pairs of the solving process and candidate answer $\{(\mathbf{s}_i, \mathbf{y}_i) \mid i = 0, 1, ..., n - 1\}$, and finally outputs its choice number $c$. However, there may not be an accurate candidate answer. Therefore, we add another choice number $n$ to the model $\mathcal{M}$. If the model $\mathcal{M}$ infers that all candidate answers are wrong, it is forced to output the choice number $n$. After the above process, the model $\mathcal{M}$ outputs its choice, as shown in Equation 3.

$$c = \mathcal{M}(\mathbf{x}_{img}; \mathbf{x}_{txt}, (\mathbf{s}_0, \mathbf{y}_0), (\mathbf{s}_1, \mathbf{y}_1), ..., (\mathbf{s}_{n-1}, \mathbf{y}_{n-1}), \mathbf{p}_{cho}), c \in \{0, 1, ..., n\}. \tag{3}$$

**Find another more promising answer.** If the model $\mathcal{M}$ infers there is no accurate candidate answer, *i.e.*, $c = n$, Self-Choose gives a prompt $\mathbf{p}_{gen}$ to guide the model $\mathcal{M}$ to find another more promising solution $\mathbf{y}_{gen}$ according to the inaccurate solving processes and candidate answers, as shown in Equation 4.

$$\mathbf{y}_{gen} = \mathcal{M}(\mathbf{x}_{img}; \mathbf{x}_{txt}, (\mathbf{s}_0, \mathbf{y}_0), (\mathbf{s}_1, \mathbf{y}_1), ..., (\mathbf{s}_{n-1}, \mathbf{y}_{n-1}), \mathbf{p}_{gen}). \tag{4}$$

Algorithm 1 provides a comprehensive summary of the procedural steps involved in Self-Choose.

---

**Algorithm 1** Self-Choose algorithm

---

**Require:** input image $\mathbf{x}_{img}$, text question $\mathbf{x}_{txt}$, model $\mathcal{M}$,
  $n$ reasoning methods $\{\mathcal{F}_0, \mathcal{F}_1, ..., \mathcal{F}_{n-1}\}$, prompts $\{\mathbf{p}_{cho}, \mathbf{p}_{gen}\}$
 1: **for** iteration $i = 0, 1, ..., n - 1$ **do**
 2:    $\mathbf{s}_i = \mathcal{M}(\mathbf{x}_{img}; \mathbf{x}_{txt}|\mathcal{F}_i)$                    ▷ Solving process (Equation 1)
 3:    $\mathbf{y}_i = \mathcal{M}(\mathbf{x}_{img}; \mathbf{x}_{txt}, \mathbf{s}_i|\mathcal{F}_i)$                    ▷ Candidate answer (Equation 2)
 4: **end for**
 5: $c = \mathcal{M}(\mathbf{x}_{img}; \mathbf{x}_{txt}, (\mathbf{s}_0, \mathbf{y}_0), ..., (\mathbf{s}_{n-1}, \mathbf{y}_{n-1}), \mathbf{p}_{cho})$          ▷ Choice number (Equation 3)
 6: **if** $c \in \{0, 1, ..., n - 1\}$ **then**
 7:    $\mathbf{y} = \mathbf{y}_c$
 8: **else**
 9:    $\mathbf{y}_{gen} = \mathcal{M}(\mathbf{x}_{img}; \mathbf{x}_{txt}, (\mathbf{s}_0, \mathbf{y}_0), ..., (\mathbf{s}_{n-1}, \mathbf{y}_{n-1}), \mathbf{p}_{gen})$   ▷ Improved answer (Equation 4)
10: **end if**
11: $\mathbf{y} = \mathbf{y}_{gen}$
12: **return y**

---

# 5 EXPERIMENTS

## 5.1 MODELS

**LLaVA-1.6-13b.** LLaVA (Liu et al., 2023b) is a powerful MLLM in the architecture that features a simple linear projection mapping visual features of the input image into a shared embedding space with the LLM language tokens. LLaVA-1.5 (Liu et al., 2023a) is an improved version of LLaVA (Liu et al., 2023b) and achieves state-of-the-art on many benchmarks. Recently, a new version, LLaVA-1.6, has been released. We use LLaVA-v1.6-vicuna-13b to test different self-correction methods.

**Gemini-Vision.** Gemini models (Team et al., 2023; Reid et al., 2024) build on top of Transformer decoders (Vaswani et al., 2017) that are enhanced with improvements in architecture and model optimization to enable stable training at scale and optimized inference on Google's Tensor Processing Units. Gemini models are operated as a black-box system, requiring the use of an application programming interface (API) to access. We test on "gemini-pro-vision" of Gemini API with default settings.

## 5.2 BENCHMARKS

**ScienceQA.** Science Question Answering (ScienceQA) (Lu et al., 2022) is a benchmark on multi-modal multiple-choice questions with diverse science topics and annotations of their answers with corresponding lectures and explanations. We use all data containing images in the test split of ScienceQA, which comprises 2017 image-question pairs.

**WHOOPS.** WHOOPS (Bitton-Guetta et al., 2023) is a benchmark for visual commonsense comprised of purposefully commonsense-defying images created by designers using publicly-available image generation tools such as Stable Diffusion (Rombach et al., 2022). This benchmark emphasizes testing MLLM's understanding and reasoning ability towards weird images. We test our method on the vision-question answering split of WHOOPS, which contains 500 images and 3362 questions. Results are evaluated with the metric BERT Matching (BEM) (Bulian et al., 2022), which approximates a reference answer to a candidate answer given a question using a language model score (Kenton & Toutanova, 2019).

**MM-Vet.** MM-Vet (Yu et al., 2023) is a benchmark that examines MLLMs on complicated multi-modal reasoning tasks. It focuses on the integration of different core vision-language capabilities, including recognition, OCR, knowledge, language generation, spatial awareness, and math. MM-Vet uses an LLM to evaluate the consistency of the MLLM responses and labeled answers, allowing MLLMs to provide open-ended responses without being constrained by specific formats. MM-Vet consists of 200 images and 218 questions. We utilize GPT-4 (Achiam et al., 2023) to evaluate the outputs of MLLMs.

## 5.3 REASONING METHODS

**IO.** Input / output (IO) Standard Prompting (Brown et al., 2020) is the standard mode of prompting. It just inputs the images and text questions and other given information to the model. The model directly outputs an answer based on the given question and available information.

**CCoT.** Compositional Chain-of-Thought (CCoT) Prompting (Mitra et al., 2023) is a zero-shot Chain-of-Thought prompting method that utilizes scene graphs to extract compositional knowledge to assist MLLM in compositional visual reasoning. Specifically, CCoT first instructs MLLM to systematically generate a scene graph of the input image in JSON format. The scene graph is requested to include three key properties of the image: the objects, their attributes, and the relationships between them. Then MLLM is prompted with the original task prompt, image and the corresponding scene graph to generate an answer. CCoT enhances the model's capability for visual understanding.

**DDCoT.** Duty-Distinct Chain-of-Thought (DDCoT) Prompting (Zheng et al., 2023) first prompts MLLM to deconstruct the input question into a sequence of basic sub-questions, breaking the complex reasoning chain into simple steps. Another LLM is forced to answer the sub-questions that can be answered without visual information while MLLM should answer the others. Finally, LLM integrates the sub-questions and sub-answers as supplementary information to give an answer to the original question. As our goal is to design a self-correct method completely depending on MLLM itself, we just prompt MLLM to generate sub-questions and simultaneously answer them, and finally give an answer according to these pieces of information related to the original question. DDCoT encourages the model to focus more on the text question and improves the model's ability of text understanding.

## 5.4 RESULTS

In this section, we present the experiment results of Self-Refine, Self-Review and Self-Choose in detail, all along with two other methods, Multi-Agent Debate (MAD) (Liang et al., 2023) and Meta-Reasoning Prompting (MRP) (Gao et al., 2024a). Self-Choose is tested on the three benchmarks, ScienceQA, Whoops, and MM-Vet, with LLaVA-1.6-13b and Gemini-Vision. Other methods are tested on the benchmark ScienceQA. They all fail to effectively self-correct reasoning.

| | | Round | | | | | |
|---|---|---|---|---|---|---|---|
| Method | Model | 0 | 1 | 2 | 3 | 4 | 5 |
| Self-Refine | LLaVA-1.6-13b | **67.63** | 63.41 | 62.96 | 63.01 | 61.87 | 62.47 |
| | Gemini-Vision | **76.20** | 51.46 | 73.08 | 55.08 | 70.25 | 56.02 |
| Self-Review | LLaVA-1.6-13b | **67.63** | 67.18 | 67.28 | 67.33 | 67.38 | 67.23 |
| | Gemini-Vision | **76.20** | 73.67 | 73.82 | 73.97 | 73.92 | 74.22 |

Table 1: Accuraies of LLaVA and Gemini on ScienceQA with Self-Refine and Self-Review.

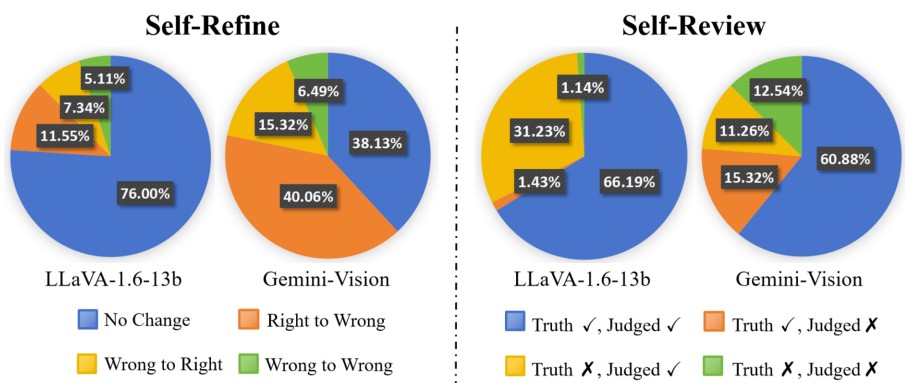

Figure 2: *Left*: Distributions of the accuracy changes in the answers of Self-Review. *No Change*: The answer remains unchanged. *Right to Wrong*: A right answer is changed to a wrong one. *Wrong to Right*: A wrong answer is changed to a right one. *Wrong to Wrong*: A wrong answer is changed but remains incorrect. *Right*: Distributions of the correctness judgment of Self-Review. *Truth ✓, Judged ✓*: An answer is right and judged as right. *Truth ✓, Judged ✗*: An answer is right but judged as wrong. *Truth ✗, Judged ✓*: An answer is wrong but judged as right. *Truth ✗, Judged ✗*: An answer is wrong and judged as wrong.

### 5.4.1 SELF-REFINE

The accuracy and the number of rounds of Self-Refine are reported in Table 1. "Round 0" represents using standard prompting without self-correction. It can be found that the model's performance drops after using Self-Refine, no matter how many rounds it takes. The reasoning accuracy does not steadily improve as the number of rounds increases. Figure 2 summarizes the results of changes in answers after a round of Self-Refine. We can take the results of LLaVA-1.6-13b as an example. It can be observed that 76% of answers remain unchanged after self-correction. While among other answers, the model is more likely to modify a correct answer to an incorrect one than to revise an incorrect answer to a correct one, resulting in the failure of Self-Refine.

### 5.4.2 SELF-REVIEW

Table 1 reports the results of Self-Review. "Rounds" in Table 1 represents the number of majority votes to judge the correctness of answers. Although Self-Review achieves overall better results than Self-Refine, it still cannot improve original answers effectively. Figure 2 summarizes the distribution of correctness in model judgment. For instance, if the original answer is actually right and the model judges it is right, or the original answer is actually wrong and the model judges it is wrong, it indicates the model makes a correct judgment. Otherwise, it is an incorrect judgment. We observe that there are 32.67% of judgments are incorrect on LLaVA and 27.86% on Gemini-Vision. This indicates that MLLMs (at least for LLaVA and Gemini) are unable to directly judge the correctness of their answers properly. Therefore, Self-Refine and Self-Review cannot effectively self-correct reasoning, and may even lead to a degradation in performance.

| Model | IO | CCoT | DDCoT | MAD-D | MAD-E | MRP |
|---|---|---|---|---|---|---|
| LLaVA-1.6-13b | **67.63** | 67.72 | 66.73 | 60.44 | 64.75 | 67.23 |
| Gemini-Vision | 76.20 | 76.40 | **78.98** | 65.84 | 69.96 | 77.39 |

Table 2: Accuracy of IO, CCoT, DDCoT, MAD-D, MAD-E, and MRP on ScienceQA, MAD-D: MAD in discriminative mode, MAD-E: MAD in extractive mode.

| Model | LLaVA-1.6-13b | | | Gemini-Vision | | |
|---|---|---|---|---|---|---|
| Benchmark | ScienceQA | WHOOPS | MM-Vet | ScienceQA | WHOOPS | MM-Vet |
| IO | 67.63 | 62.20 | 46.16±0.14 | 76.20 | 68.34 | 58.80±0.37 |
| CCoT | 67.72 | 62.53 | 47.78±0.22 | 76.40 | 68.01 | 60.56±0.27 |
| DDCoT | 66.73 | 60.89 | 41.66±0.42 | 78.98 | 63.08 | 57.72±0.21 |
| IO - SC | 67.87 | - | - | 76.85 | - | - |
| CCoT - SC | 68.42 | - | - | 76.80 | - | - |
| DDCoT - SC | 68.12 | - | - | 79.77 | - | - |
| Self-Choose | **68.86** | **62.65** | **48.28±0.22** | **80.02** | **69.12** | **62.84±0.19** |

Table 3: Main results of different reasoning methods and Self-Choose on ScienceQA, WHOOPS, MM-Vet benchmarks. We use GPT-4 to evaluate the results on MM-Vet five times, and show GPT-4 Score in the form of *"mean ± standard deviation"*. The best result is in **bold**. More details can be found in Appendix D.

### 5.4.3 MAD AND MRP

MAD sets the LLM instances to play different roles as affirmative and negative sides to debate with each other, which can alleviate the issue of *self-reflection* in LLMs. A judge model is assigned to determine the final solution. In the discriminative mode, the judge chooses the side it supports. In the extractive mode, it summarizes their opinions to give a final answer. MRP is an approach similar to ours, which uses a long system prompt to guide LLMs to first choose the most suitable prompting method, and then solve the problem. We set MRP to choose from the same 3 candidate methods as Self-Choose to make a fair comparison.

Table 2 reports the accuracy of different basic prompting methods, along with MAD and MRP. Experiment results show that MAD gets worse performance both on LLaVA and Gemini. On LLaVA, MRP performs worse than IO. On Gemini, it achieves better performance than IO while worse than DDCoT, which is consistent with its experiments on LLMs (Gao et al., 2024a). It indicates that MRP can not choose the most optimal method before solving the problem. Otherwise, it would achieve better performance than all candidate methods.

### 5.4.4 SELF-CHOOSE

We compare Self-Choose with IO, CCoT, DDCoT, and these with Self-Consistency (SC) (Wang et al., 2023). SC calls the model three times to vote for the most repeated answer. We do not test SC on WHOOPS and MM-Vet, as they are open-ended Q&A but SC is only applicable to problems where the final results are numbers, options, *etc*. Experiment results of LLaVA-1.6-13b and Gemini-Vision in different methods are shown in Table 3. The performance of these three reasoning methods varies across different benchmarks and metrics. Taking the results of Gemini-Vision as an example, CCoT outperforms IO and DDCoT on MM-Vet, while DDCoT achieves the highest accuracy among the three reasoning methods on ScienceQA. Nonetheless, no matter which reasoning method is employed, the performance is enhanced after Self-Choose. Our method also outperforms SC and can be applied to various open-ended Q&A scenarios. This proves the effectiveness of our method compared to other self-correction methods, which can compare different reasoning solutions and choose the optimal one, as the example shown in Figure 3.

### 5.5 ABLATION STUDY

We perform a comprehensive ablation study on LLaVA. Detailed results are shown in Table 4 and 5.

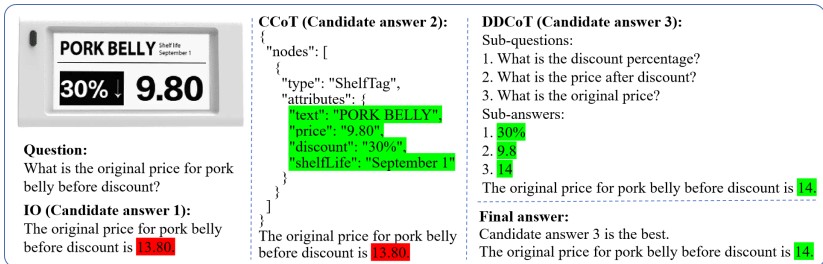

Figure 3: Example of Self-Choose. IO and CCoT generate wrong answers, while the solving process of CCoT is correct. DDCoT provides the correct solving process and answer. By reflecting and comparing these solving processes and candidate answers, Self-Choose chooses the right candidate and outputs the correct answer.

| Benchmark | ScienceQA | WHOOPS | MM-Vet |
|---|---|---|---|
| All IO | 67.87 | 61.97 | 46.18±0.04 |
| All CCoT | 68.27 | 62.47 | **48.46±0.29** |
| All DDCoT | 67.33 | 61.5 | 39.68±0.19 |
| w/o choice $n$ | 67.87 | 62.43 | 46.94±0.37 |
| w/o processes $\mathbf{s}_i$ | 68.81 | 62.00 | 43.52±0.24 |
| Generate | 66.78 | 55.98 | 41.16±0.29 |
| Self-Choose | **68.86** | **62.65** | 48.28±0.22 |

Table 4: Results of the ablation study. *All IO*: Replace all three reasoning methods with IO. *All CCoT*: Replace all three reasoning methods with CCoT. *All DDCoT*: Replace all three reasoning methods with DDCoT. *w/o choice* $n$: Remove the choice number $n$. *w/o processes* $\mathbf{s}_i$: Remove the solving processes $\mathbf{s}_i$ in Equation 3 and 4. *Generate*: Generate an answer without choosing the best candidate answer. The best result is in **bold** and the suboptimal is underlined.

**Replace the three reasoning methods with the same one.** In this ablation study, we replace the three reasoning method with the same one on the model LLaVA-1.6-13b, *i.e.*, covert $(\mathcal{F}_0, \mathcal{F}_1, \mathcal{F}_2)$ to $(\mathcal{F}_j, \mathcal{F}_j, \mathcal{F}_j \mid j \in \{0, 1, 2\})$. In the same way, the model outputs the choice number of the best answer among the three candidate answers generated in the same reasoning method. If none of candidate answers is accurate, the model is forced to generate a more promising answer according to these inaccurate candidate answers. Self-Choose achieves the best performance on ScienceQA, WHOOPS and the highest BEM on MM-Vet, which proves that Self-Choose can effectively improve answers through reflecting and comparing the solving processes and candidate answers of different reasoning methods. What's more, the model is forced to output solving processes with long tokens when using complex reasoning methods. Concatenating solving processes as context information also introduces more token consumption. Therefore, Self-Choose is more efficient than replacing different reasoning methods with the same one.

**Remove the choice number** $n$**.** We test the performance that the model only chooses the best candidate answer, without generating a more promising answer if candidate answers are all inaccurate, *i.e.*, removing the choice number $n$. It gets worse results compared to Self-Choose. The results indicate that it is necessary to incorporate the choice number $n$ to generate a more promising answer if all candidate answers are deemed inaccurate. This offers a chance to improve answers when all reasoning methods fail to produce right answers.

**Remove the solving processes** $\mathbf{s}_i$**.** We conduct experiments to verify the necessity of solving processes $\mathbf{s}_i$ when choosing the best candidate answer. Without solving processes, Self-Choose fails to choose the best candidate and gets worse performance.

**Generate an answer without choosing the best candidate answer.** We evaluate the performance that the model only generates an improved answer according to candidate answers without choosing the best candidate answer, *i.e.*, executing Equation 4 by replacing $\mathbf{p}_{gen}$ with another prompt $\mathbf{p}^*_{gen}$. $\mathbf{p}_{gen}$ tells the model all candidate answers are inaccurate while $\mathbf{p}^*_{gen}$ does not. The performance mainly drops down in this setting. We observe that the model trends to generate an answer different

|          | Setting |       |       |       |       |
| :------: | :-----: | :---: | :---: | :---: | :---: |
| Round $N$ |   (1)   |  (2)  |  (3)  |  (4)  |  (5)  |
|    1     |  68.86  | 68.72 | **68.77** | **68.77** |   -   |
|    3     |  68.86  | **68.77** | **68.77** | 68,72 | 68.72 |
|    5     | **68.96** | **68.77** | **68.77** | **68.77** |   -   |

Table 5: Accuracy of LLaVA on ScienceQA with different settings and rounds.

from candidate answers. However, in the majority of cases, it exists the correct answer in candidate answers. Therefore, the strategy of Self-Choose is better as it forces the model to generate an improved answer only when all candidate answers are judged as wrong.

**Other settings for Equation 4.** we design five different settings to further analyze the last step to generate a more promising answer (Equation 4). Here are the specific settings:

- (1) Generate a more promising answer $N$ times by reviewing wrong candidate answers, then choose the best one among them.

- (2) Divide the last step into two steps. Find the problems of wrong candidate answers at first, then generate a more promising answer according to the problems. This will be repeated $N$ times, then choose the best promising answer.

- (3) Change the prompt $\mathbf{p}_{gen}$ to CCoT. Generate a scene graph at first by reviewing wrong candidate answers, then answer the question according to the scene graph. This will be repeated $N$ times, then choose the best one.

- (4) Change the prompt $\mathbf{p}_{gen}$ to DDCoT. Deconstruct the question down to sub-questions and get sub-answers by reviewing wrong candidate answers, then answer the question according to sub-questions and sub-answers. This will be repeated $N$ times, then choose the best one.

- (5) Generate a more promising answer with the original prompt $\mathbf{p}_{gen}$, CCoT and DDCoT, respectively. Replace candidate answers with them and repeat the process of Self-Debate, until it succeeds to choose the best candidate answer. We set the maximum number of rounds to $N$, and force the model to choose the best one at the maximum round.

Table 5 summarizes the results with different settings and rounds. It shows that there is no need to design complex instructions for the last step, the original prompt $\mathbf{p}_{gen}$ in our paper is the best setting. This may have similar reasons to the failure of Self-Refine. What's more, the performance will gain minor improvement as the round $N$ increases.

**Extend Self-Choose to natural language domain.** Our method is initially designed for MLLMs, which utilizes diverse reasoning solutions focusing on different aspects such as image understanding and text understanding. However, its core idea can also be applied to natural language domain. We extent experiments on LLMs, and surprisingly find Self-Choose can also successfully help LLMs self-correct reasoning. Please refer to Appendix E for more details.

## 6 CONCLUSION

We propose Self-Choose: an effective prompting approach to teach MLLMs to self-correct like humans. Our approach entirely relies on a single model to correct reasoning, without the assistance of any additional tools or models, and does not require training or fine-tuning. Self-Choose compares the reasoning processes and outcomes of different reasoning methods to select the best answer or generate improved solutions based on the processes and results of various reasoning methods. Experiments on three reasoning benchmarks implemented on LLaVA-1.6-13b and Gemini-Vision demonstrate that our method can truly and effectively self-correct reasoning. We hope that our work will provide new insights into self-correction on reasoning and foster research in this area.

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

## A    SOCIAL IMPACT

We propose an effective prompting strategy, Self-Choose, to self-correct reasoning without external feedback, which can be applied to black-box MLLMs. Our research can facilitate the exploration of the potential of MLLMs, leveraging the models' intrinsic capabilities for self-correction and self-improvement. It is significant to study self-correction because we cannot always rely on stronger models to help with correction. For example, how can we find a more powerful model to correct the strongest model? A feasible strategy is to introduce human supervision to assist models in correcting errors. Nevertheless, this may be time-consuming and laborious. What's more, can we still effectively supervise when the model is stronger than humans? So how can we correct the most powerful model? An intuitive approach is to teach the models to self-correct. We hope that our research will provide insights into reasoning self-correction and stimulate further research in this area. What's more, our approach could also be replicated and applied to LLMs, potentially enhancing their capacity for self-correction. However, it is difficult to guard against the potential misuse of this technology by malefactors for illicit activities.

## B    LIMITATIONS

The main limitation of Self-Choose is that the base model should have a certain level of reasoning ability. If the model's reasoning capabilities are weak, with extensive errors across a multitude of tasks and reasoning methods, then it will be challenging to enhance its performance using Self-Choose. Although our method is capable of effectively self-correcting reasoning, it occasionally falls short, as demonstrated by the failure case illustrated in Figure 14. This may be because, although this method is capable of correcting reasoning errors, it may not be effective for issues related to the model's cognitive limitations. For instance, if the model is not aware of the existence of the platypus and mistakenly identifies it as a duck, it cannot rectify its understanding to recognize the animal as such through self-correction.

## C    IMPLEMENTATION DETAILS

In our experiments, we set the temperature of LLaVA-1.6-13b to 0.3, which encourages the model to generate answers with relatively high certainty, while still ensuring a level of diversity. We run the model on two Tesla-V100 GPUs. When testing each method on the benchmark ScienceQA, we prompt the MLLM to also provide the rationale behind its choices instead of just forcing the MLLM to output the option only. Self-Choose can more effectively reflect on the responses of each method, as it provides reasons rather than isolated options. The prompt is *"Only one option is correct. Please choose the right option and explain why you choose it. You must answer in the following format. For example, if the right answer is A, you should answer: The answer is A. Because ...".*

Due to the inherent mechanisms of MLLM, the output of the model $\mathcal{M}$ may contain nonsensical sentences or not conform to the stipulated format. To mitigate this issue, we set up some templates to extract the choice number in the model response, which is noted as $\varphi$. If the model output contains nonsensical sentences or is not in the stipulated format, $\varphi$ will return $None$. We implement a repetitive generation process, continuing until the choice number is successfully extracted or the iteration count exceeds a predetermined threshold $T$. If the number of iterations reaches $T$ but $\varphi(c) = None$, then sample an element at random from the set $\{0, 1, ..., n\}$ and assign it to $c$.

To facilitate comprehension, in Section 5, we designate $\mathcal{F}_0$ as the representative of standard prompting. However, in the design of our prompts, we utilize the numerical notation from 1 to $n$ to denote the reasoning processes and answers of various reasoning methods. This is more in tune with the conventions of human communication and the structure of internet text.

## D MORE DETAILED EXPERIMENT RESULTS

Figure 4 summarizes the distributions of the accuracy changes in the answers of Self-Choose on the benchmark ScienceQA. The majority retains the original answers. Compared to the number of right answers that are incorrectly altered to a wrong one, more wrong answers are corrected to the right ones. For example, 1.54% of original answers of LLaVA-1.6-13b are incorrectly changed from right to wrong, while 2.73% of original answers are correctly changed from wrong to right. For Gemini-Vision, 0.55% of original answers are incorrectly changed from right to wrong, while 4.36% are correctly changed from wrong to right. With more wrong answers being properly corrected, the reasoning performance is improved.

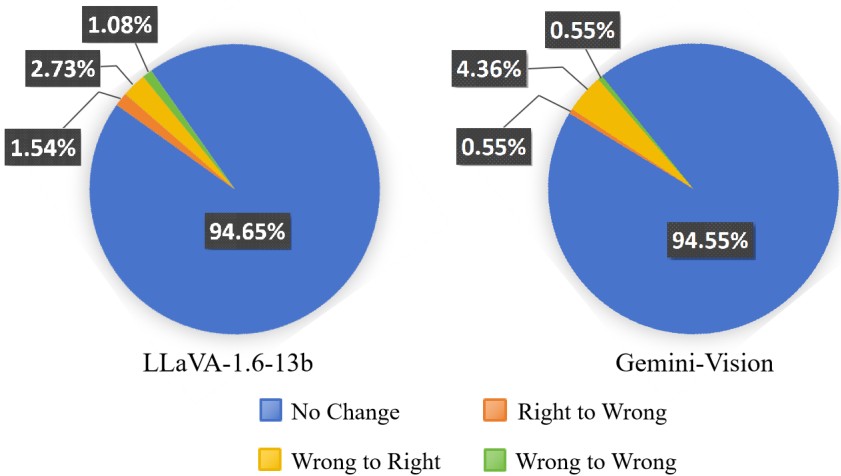

Figure 4: Distributions of the accuracy changes in the answers of Self-Choose on the benchmark ScienceQA. *No Change*: The answer remains unchanged. *Right to Wrong*: A right answer is changed to a wrong one. *Wrong to Right*: A wrong answer is changed to a right one. *Wrong to Wrong*: A wrong answer is changed but remains incorrect.

Table 6 and 7 summarize the accuracy of results on the benchmark ScienceQA with LLaVA-1.6-13b and Gemini-Vision, respectively. Self-Choose performs well in the three aspects, especially in natural science. Self-Choose achieves the highest accuracy in natural science and social science, performing the best overall.

|  | Total | Natural Science | Social Science | Language Science |
|---|---|---|---|---|
| IO | 67.63 (1364 / 2017) | 66.58 (805 / 1209) | 68.72 (525 / 764) | 77.27 (34 / 44) |
| CCoT | 67.72 (1366 / 2017) | 65.84 (796 / 1209) | **69.90** (534 / 764) | **81.81** (36 / 44) |
| DDCoT | 66.73 (1346 / 2017) | 66.67 (806 / 1209) | 66.23 (506 / 764) | 77.27 (34 / 44) |
| Self-Choose | **68.86** (1389 / 2017) | **67.91** (821 / 1209) | **69.90** (534 / 764) | 77.27 (34 / 44) |

Table 6: Detailed results on the benchmark ScienceQA with LLaVA-1.6-13b.

|  | Total | Natural Science | Social Science | Language Science |
|---|---|---|---|---|
| IO | 76.20 (1537 / 2017) | 70.05 (847 / 1209) | 85.34 (652 / 764) | 86.36 (38 / 44) |
| CCoT | 76.40 (1541 / 2017) | 69.98 (846 / 1209) | 85.86 (656 / 764) | 88.64 (39 / 44) |
| DDCoT | 78.98 (1593 / 2017) | 73.37 (887 / 1209) | **86.91** (664 / 764) | **95.45** (42 / 44) |
| Self-Choose | **80.02** (1614 / 2017) | **75.27** (910 / 1209) | **86.91** (664 / 764) | 90.90 (40 / 44) |

Table 7: Detailed results on the benchmark ScienceQA with Gemini-Vision.

Table 8 and 9 summarize the GPT-4 Score in the six parts of MM-Vet with LLaVA-1.6-13b and Gemini-Vision, respectively. Our method improves the performance in multiple aspects compared with other reasoning methods. Specifically, Self-Choose with Gemini-Vision achieves the best GPT-4 Score on *OCR*, *Know*, *Gen*, *Spat* and the suboptimal GPT-4 Score on *Math* and *Rec*, performing the best in total. For each question, some reasoning methods may provide incorrect solutions, while others may generate correct ones. Self-Choose selects the most likely correct answer by comparing the solving processes and answers of these methods, thus enhancing the performance compared to each method.

| | Total | OCR | Know | Gen | Spat | Math | Rec |
|---|---|---|---|---|---|---|---|
| IO | 46.16±0.14 | 42.74±0.30 | 37.60±0.47 | 40.66±0.29 | 43.32±0.34 | 26.50±0.00 | 48.80±0.19 |
| CCoT | 47.78±0.22 | 43.94±0.36 | **39.8±0.65** | **43.04±0.55** | 44.38±0.44 | **30.40±00** | 50.26±0.31 |
| DDCoT | 41.66±0.42 | 38.04±0.61 | 31.36±0.81 | 33.68±0.90 | 41.50±0.24 | 30.08±0.16 | 43.24±0.47 |
| Self-Choose | **48.28±0.22** | **45.34±0.30** | 39.04±0.24 | 42.48±0.50 | **48.28±0.34** | 26.50±0.00 | **50.98±0.33** |

Table 8: Details of GPT-4 Score on the benchmark MM-Vet with LLaVA-1.6-13b. *OCR*: Optical character recognition. *Know*: Knowledge. Vision-question answering that covers various knowledge-related capabilities, including social and visual commonsense knowledge. *Gen*: Language generation. *Spat*: Spatial awareness. *Math*: Written equations or problems in the wild. *Rec*: Visual recognition.

| | Total | OCR | Know | Gen | Spat | Math | Rec |
|---|---|---|---|---|---|---|---|
| IO | 58.80±0.37 | 55.44±0.45 | 63.40±0.71 | 42.54±0.48 | 38.42±0.90 | **65.48±0.44** | 54.76±2.74 |
| CCoT | 60.56±0.27 | 58.80±0.30 | 62.58±0.49 | 48.24±0.44 | 44.58±0.46 | 65.04±0.17 | 56.78±1.79 |
| DDCoT | 57.72±0.21 | 54.82±0.40 | 59.32±0.52 | 45.76±0.34 | 41.64±0.45 | 60.56±0.52 | **58.80±0.00** |
| Self-Choose | **62.84±0.19** | **61.26±0.05** | **64.48±0.46** | **51.50±0.19** | **48.08±0.20** | 65.06±0.14 | 57.32±1.56 |

Table 9: Details of GPT-4 Score on the benchmark MM-Vet with Gemini-Vision.

# E EXPERIMENTS ON NATURAL LANGUAGE DOMAIN

We extend experiments with GPT-4o-mini-2024-07-18 (Open AI, 2024) on the test split of GSM8K (Cobbe et al., 2021), which contains diverse grade school math problems. We randomly sample 250 samples, and adopt three reasoning methods: IO (just output the result), Chain-of-Thought (Wei et al., 2022), and Least-to-Most (Zhou et al., 2023). Self-Choose outperforms other methods, as shown in Table 10.

| Model | IO | CoT | Least-to-Most | Self-Choose |
|---|---|---|---|---|
| GPT-4o-mini | 31.2 | 92.8 | 93.2 | 94.8 |

Table 10: Accuracy of each method on GSM8K.

What's more, we test 5 high school math problems on the website of MathGPT (Tomorrow Advancing Life, 2023), which Self-Refine and Self-Review all fail to self-correct. Our method succeeds to correct original answers in all problems. Figure 5 shows an example. These demonstrate the superiority and generality of our method. We believe that our method can be widely applied in more scenarios.

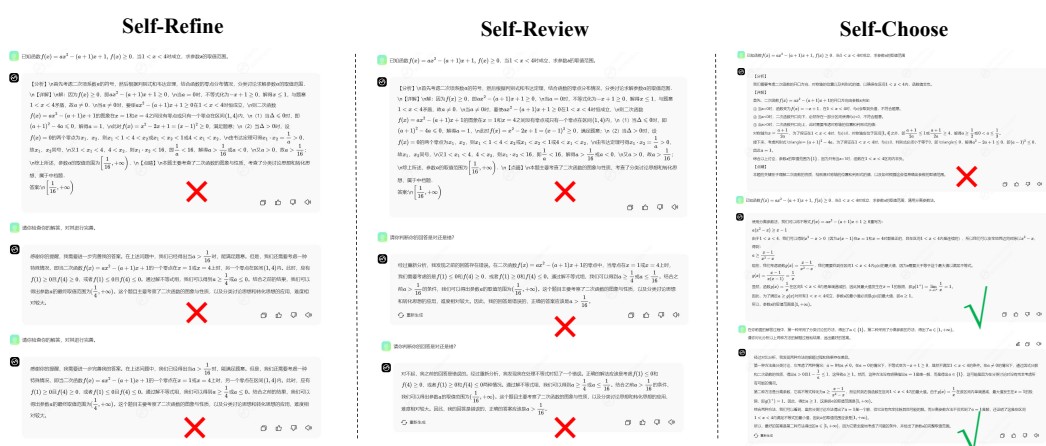

Figure 5: Example on MathGPT.

## F  EXAMPLES IN SELF-REFINE

**Question:**
Which type of force from the people moves the car forward?

**Options:**
A. push          B. pull

---

**Answer:**
The answer is A. Because the people in the image are pushing the car out of the snow. They are not pulling it. Pushing is the force that moves the car forward in this situation, as they are applying pressure from the back of the car towards the front, which is the opposite direction of the direction they want the car to move.

**Find problems (Round 1):**
Upon reviewing the image, I realize that my previous answer was incorrect. The people in the image are indeed pulling the car out of the snow, not pushing it. They are applying a force from the front of the car towards the back, which is the direction they want the car to move. Therefore, the correct answer is B. Pull. I apologize for the mistake in my previous response.

**Revised answer (Round 1):**
The answer is B. Pull. Because the people in the image are pulling the car out of the snow, not pushing it. They are applying a force from the front of the car towards the back, which is the direction they want the car to move. This is the opposite direction of pushing, which would be from the back of the car towards the front. The correct force applied in this situation is pulling, as the people are working together to move the car in the desired direction.

**Find problems (Round 2):**
The image shows three individuals working together to move a car out of the snow. They are applying a force from the front of the car towards the back. This is the direction they want the car to move. The force applied in this situation is pulling, as they are working together to move the car in the desired direction. The correct answer is B. Pull.

**Revised answer (Round 2):**
The answer is B. Pull.
Because the people in the image are pulling the car out of the snow, not pushing it. They are applying a force from the front of the car towards the back, which is the direction they want the car to move. This is the opposite direction of pushing, which would be from the back of the car towards the front. The correct force applied in this situation is pulling, as the people are working together to move the car in the desired direction.

**Find problems (Round 3):**
Upon reviewing the image and the context provided, I realize that my previous answer was incorrect. The people in the image are indeed pushing the car out of the snow, not pulling it. They are applying a force from the back of the car towards the front, which is the direction they want the car to move. This is the opposite direction of pulling, which would be from the front of the car towards the back. The correct force applied in this situation is pushing, as the people are working together to move the car in the desired direction.
I apologize for the mistake in my previous answer.

**Revised answer (Round 3):**
The answer is A. Push.
Because the people in the image are pushing the car out of the snow, not pulling it. They are applying a force from the back of the car towards the front, which is the direction they want the car to move. This is the opposite direction of pulling, which would be from the front of the car towards the back. The correct force applied in this situation is pushing, as the people are working together to move the car in the desired direction.

Figure 6: Example of Self-Refine on the benchmark ScienceQA.

**Question:**
Which property do these two objects have in common?

**Options:**
A. sour          B. stretchy

**Answer:**
The answer is B. Because both objects have the ability to stretch.

**Find problems (Round 1):**
The answer is not B. The correct answer should be (A) sour.
Because both objects have the ability to be sour.

**Revised answer (Round 1):**
The answer is A. Because both objects have the ability to be sour.

**Find problems (Round 2):**
The answer is B. The two objects have the property of being stretchy in common. A balloon can be stretched by blowing air into it, and a spring can be stretched by pulling on it.

**Revised answer (Round 2):**
The answer is B. Because both objects have the ability to be stretchy. A balloon can be stretched by blowing air into it, and a spring can be stretched by pulling on it.

**Find problems (Round 3):**
The answer should be (A). The two objects in the picture are a balloon and a spring. A balloon is not sour, but a spring can be made of steel which can rust and taste sour. Therefore, the correct answer is (A) sour.

**Revised answer (Round 3):**
The answer is A. Because a balloon is not sour, but a spring can be made of steel which can rust and taste sour.

**Find problems (Round 4):**
The answer should be (B).
The common property of these two objects is that they are both stretchy. A balloon can be stretched by blowing air into it, and a spring can be stretched by pulling on it.

**Revised answer (Round 4):**
The answer is B.
Because a balloon can be stretched by blowing air into it, and a spring can be stretched by pulling on it.

**Find problems (Round 5):**
The answer is not B.
The problem with the previous answer is that the two objects are not stretchy in the same way. A balloon is stretchy because it is made of a thin, elastic material. A spring is stretchy because it is made of a coiled metal. The two objects are not interchangeable.

**Revised answer (Round 5):**
The answer is B.
Because a balloon can be stretched by blowing air into it, and a spring can be stretched by pulling on it.

Figure 7: Example of Self-Refine on the benchmark ScienceQA.

## G    EXAMPLES IN SELF-REVIEW

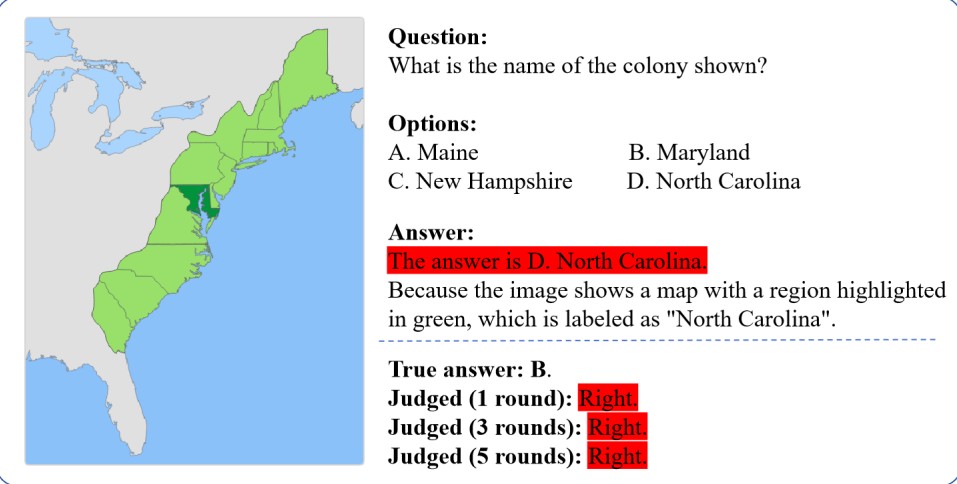

Figure 8: Example of Self-Review on the benchmark ScienceQA.

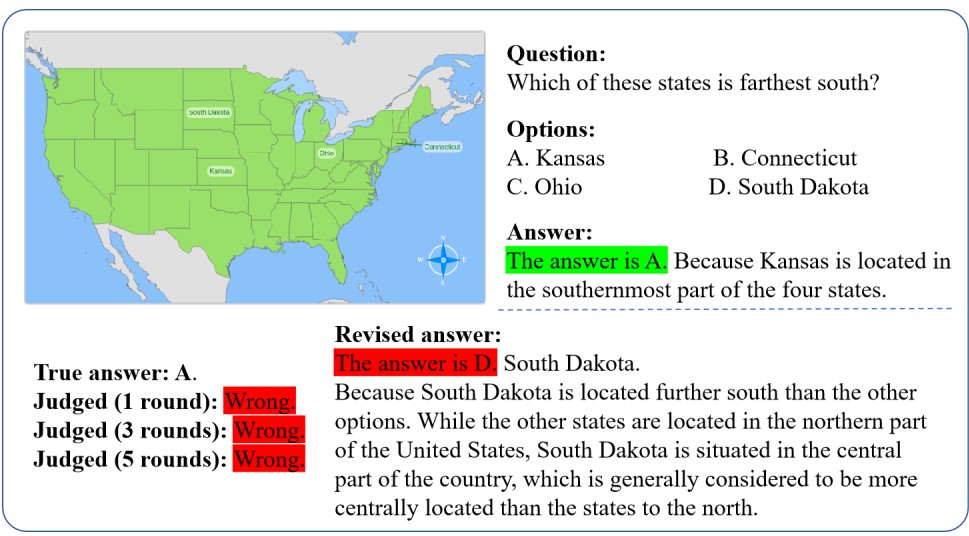

Figure 9: Example of Self-Review on the benchmark ScienceQA.

**Question:**
Which solution has a higher concentration of green particles?

**Options:**
A. Solution B
B. neither; their concentrations are the same
C. Solution A

Solvent volume: 45 mL      Solvent volume: 45 mL
Solution A                 Solution B

**Answer:**
The answer is B. Because both solutions have the same amount of solute (green particles) and the same amount of solvent (45 mL). Therefore, their concentrations are the same.

**True Answer: A**
**Judged (1 round):** Wrong.
**Judged (3 rounds):** Wrong.
**Judged (5 rounds):** Right.

**Revised answer:**
The answer is B. Because the concentration of green particles in both Solution A and Solution B is 45 mL, which means they have the same concentration of green particles. Therefore, the concentration is the same in both solutions.

Figure 10: Example of Self-Review on the benchmark ScienceQA.

## H  EXAMPLES OF SELF-CHOOSE

**Question:**
Which solution has a higher concentration of green particles?

**Options:**
A. Solution B
B. neither; their concentrations are the same
C. Solution A

Solvent volume: 45 mL        Solvent volume: 45 mL
Solution A                   Solution B

**IO (Candidate answer 1):**
The answer is B. Because both solutions have the same amount of solute (green particles) and the same amount of solvent (45 mL). Therefore, their concentrations are the same.

**CCoT (Candidate answer 2):**
```
{
  "nodes": [
    {
      "id": "Solution A",
      "attributes": {
        "solvent volume": 45,
        "solute particle count": 8
      }
    },
    {
      "id": "Solution B",
      "attributes": {
        "solvent volume": 45,
        "solute particle count": 10
      }
    }
  ],
  "edges": [
    {
      "source": "Solution A",
      "target": "Solution B",
      "type": "concentration
              comparison",
      "attributes": {
        "difference": 2
      }
    }
  ]
}
```
The answer is B.

**DDCoT (Candidate answer 3):**

Sub-questions:
1. What is the definition of concentration?
2. What information do we have about the two solutions?
3. What can we conclude from the information we have?

Sub-answers:
1. Concentration is the amount of solute dissolved in a given volume of solvent.
2. We know that both solutions have the same volume of solvent (45 mL) and that Solution A has 12 green particles while Solution B has 15 green particles.
3. We can conclude that Solution B has a higher concentration of green particles than Solution A.

The answer is A, Solution B.

**Final answer:**
The candidate answer 3 is the best.
The answer is A.

Figure 11: Example of Self-Choose on the benchmark ScienceQA.

**Question:**
What part of a tree are potatoes growing on?

**IO (Candidate answer 1):**
Potatoes do not grow on trees.

**DDCoT (Candidate answer 3):**
Sub-questions:
1. Are potatoes fruits or vegetables?
2. What part of a plant do potatoes grow on?
3. Can potatoes grow on trees?

Sub-answers:
1. Potatoes are vegetables.
2. Potatoes grow on the roots of a plant.
3. No, potatoes cannot grow on trees.

Potatoes cannot grow on trees.

**CCoT (Candidate answer 2):**

```
{
 "nodes": [
   {
     "id": "tree",
     "type": "tree",
     "attributes": {
       "species": "oak",
       "age": 100
     }
   },
   {
     "id": "potatoes",
     "type": "vegetable",
     "attributes": {
       "species": "potato",
       "count": 10
     }
   },
   {
     "id": "ground",
     "type": "ground",
     "attributes": {
       "type": "soil"
     }
   }
 ],
 "edges": [
   {
     "source": "potatoes",
     "target": "tree",
     "type": "attached_to"
   },
   {
     "source": "tree",
     "target": "ground",
     "type": "rooted_in"
   }
 ]
}
```

Potatoes do not grow on trees.

**Final answer:**
The candidate answers are all wrong.
A more promising answer is:
The image shows potatoes growing on the trunk of a tree. This is impossible because potatoes cannot grow on trees. The image is likely a digitally generated artwork.

Figure 12: Example of Self-Choose on the benchmark WHOOPS.

**Question:**
What is the number displayed on the motorcycle ridden by the motorcyclist wearing a yellow suit?

**IO (Candidate answer 1):**
16.

**DDCoT (Candidate answer 3):**
Sub-questions:
1. What is the number displayed on the motorcycle ridden by the motorcyclist wearing a yellow and blue suit?
2. What is the number displayed on the motorcycle ridden by the motorcyclist wearing a yellow suit?

Sub-answers:
1. 17
2. 9

17 and 9.

**Final answer:**
Candidate answer 1 is the best.
16.

**CCoT (Candidate answer 2):**
```
{
  "objects": [
    {
      "type": "motorcycle",
      "attributes": {
        "color": "yellow",
        "number": 16
      }
    },
    {
      "type": "person",
      "attributes": {
        "suit_color": "yellow",
        "helmet_color": "black"
      }
    }
  ],
  "relationships": [
    {
      "type": "drives",
      "subject": {
        "type": "person",
        "attributes": {
          "suit_color": "yellow",
          "helmet_color": "black"
        }
      },
      "object": {
        "type": "motorcycle",
        "attributes": {
          "color": "yellow",
          "number": 16
        }
      }
    }
  ]
}
```
16.

Figure 13: Example of Self-Choose on the benchmark MM-Vet.

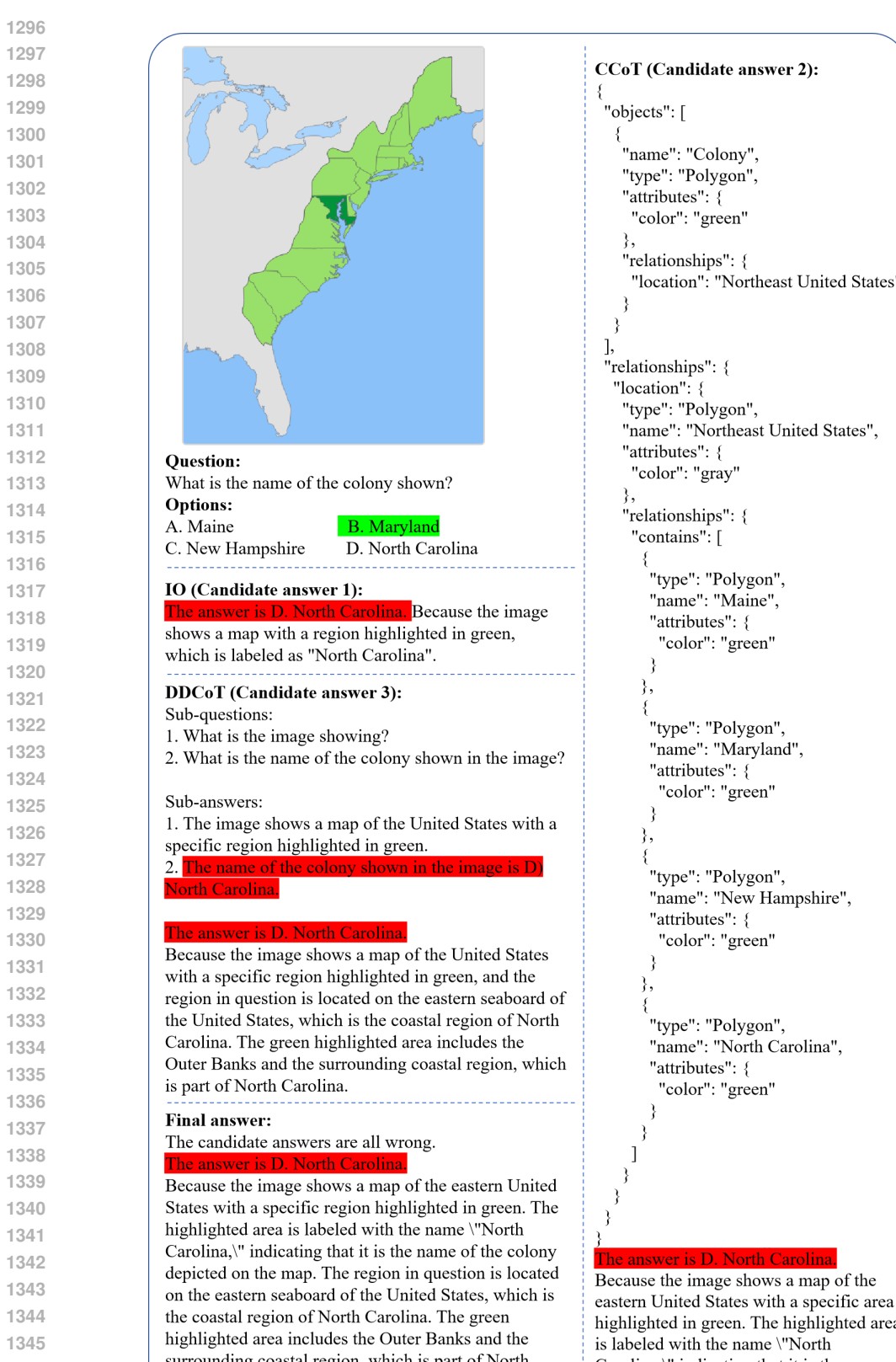

**Question:**
What is the name of the colony shown?
**Options:**
A. Maine          B. Maryland
C. New Hampshire          D. North Carolina

**IO (Candidate answer 1):**
The answer is D. North Carolina. Because the image shows a map with a region highlighted in green, which is labeled as "North Carolina".

**DDCoT (Candidate answer 3):**
Sub-questions:
1. What is the image showing?
2. What is the name of the colony shown in the image?

Sub-answers:
1. The image shows a map of the United States with a specific region highlighted in green.
2. The name of the colony shown in the image is D) North Carolina.

The answer is D. North Carolina.
Because the image shows a map of the United States with a specific region highlighted in green, and the region in question is located on the eastern seaboard of the United States, which is the coastal region of North Carolina. The green highlighted area includes the Outer Banks and the surrounding coastal region, which is part of North Carolina.

**Final answer:**
The candidate answers are all wrong.
The answer is D. North Carolina.
Because the image shows a map of the eastern United States with a specific region highlighted in green. The highlighted area is labeled with the name \"North Carolina,\" indicating that it is the name of the colony depicted on the map. The region in question is located on the eastern seaboard of the United States, which is the coastal region of North Carolina. The green highlighted area includes the Outer Banks and the surrounding coastal region, which is part of North Carolina.

**CCoT (Candidate answer 2):**

```
{
  "objects": [
    {
      "name": "Colony",
      "type": "Polygon",
      "attributes": {
        "color": "green"
      },
      "relationships": {
        "location": "Northeast United States"
      }
    }
  ],
  "relationships": {
    "location": {
      "type": "Polygon",
      "name": "Northeast United States",
      "attributes": {
        "color": "gray"
      },
      "relationships": {
        "contains": [
          {
            "type": "Polygon",
            "name": "Maine",
            "attributes": {
              "color": "green"
            }
          },
          {
            "type": "Polygon",
            "name": "Maryland",
            "attributes": {
              "color": "green"
            }
          },
          {
            "type": "Polygon",
            "name": "New Hampshire",
            "attributes": {
              "color": "green"
            }
          },
          {
            "type": "Polygon",
            "name": "North Carolina",
            "attributes": {
              "color": "green"
            }
          }
        ]
      }
    }
  }
}
```

The answer is D. North Carolina.
Because the image shows a map of the eastern United States with a specific area highlighted in green. The highlighted area is labeled with the name \"North Carolina,\" indicating that it is the name of the colony depicted on the map.

Figure 14: Failure case of Self-Choose on the benchmark ScienceQA.

# I  PROMPTS

In this section, we show the prompts of Self-Refine, Self-Review, CCoT, DDCoT, Self-Choose, and prompts using in Section 5.5. "***Answer:***" represents the response of the MLLM.

Image:
<image>

Question:
<question>

***Answer:***
*<answer>*

---

Review your previous answer and find problems with your answer.

***Answer:***
*<problems>*

---

Based on the problems of your previous answer, improve your answer.

***Answer:***
*<revised answer>*

Figure 15: Prompts of Self-Refine.

Image:
<image>

Question:
<question>

***Answer:***
*<answer>*

---

Review your previous answer, determine whether it is right or wrong. You must only answer "right" or "wrong" directly. Do not say any other words.

***Answer:***
*<judgement>*

Figure 16: Prompts of Self-Review.

Image:
<image>

Question:
<question>

For the provided image and its associated question, generate a scene graph in JSON format that includes the following:
1. Objects that are relevant to answering the question.
2. Object attributes that are relevant to answering the question.
3. Obect relationships that are relevant to answering the question.

Just generate the scene graph in JSON format. Do not say extra words.

*Answer:*
*<scene graph>*

- - - - - - - - - - - - - - - - - - - - - - - - - - - - - - - - - - - - - - - - - - - -

Use the image and scene graph as context and answer the following question.
<question>

*Answer:*
*<answer>*

Figure 17: Prompts of CCoT.

Image:
<image>

Question:
<question>

Given the image and question, please think step-by-step about the preliminary knowledge to answer the question, deconstruct the problem as completely as possible down to necessary sub-questions. Then with the aim of helping humans answer the original question, try to answer the sub-questions. The expected answering form is as follows:
Sub-questions:
1. <sub-question 1>
2. <sub-question 2>
...

Sub-answers:
1. <sub-answer 1>
2. <sub-answer 2>
...

**Answers:**
*<sub-questions>*
*<sub-answers>*
- - - - - - - - - - - - - - - - - - - - - - - - - - - - - - - - - - - - - - - - - - - - - - - - -
Give your answer of the question according to the sub-questions and sub-answers.

**Answers:**
*<answer>*

Figure 18: Prompts of DDCoT.

Image:
<image>

Question:
<question>

Here are some candidate answers using different methods.
1. [
Directly answer the question.
<answer1>
]

2. [
First, get the scene graph of the image in JSON format:
<scene_graph>

Then, use the image and scene graph as context to answer the question.
<answer2>
]

3. [
First, the problem can be deconstructed down to sub-questions.
<sub-questions>
<sub-answers>

Then, according to the sub-questions and sub-answers to answer the question.
<answer3>
]

Compare these candidate answers and their solving processes to reflect. Please choose the best candidate answer. You should only answer the number (1, 2 or 3) of candidate answers. If all the candidate answers above are incorrect, you should answer the number "4" only.

*Answer:*
*<choice>*

- - - - - - - - - - - - - - - - - - - - - - - - - - - - - - - - - - - - - - - - - - - - - -

These candidate answers are all wrong. Find the problems of them, and generate a more promising answer according to these candiate answers.

*Answer:*
*<more promising answer>*

Figure 19: Prompts of Self-Choose.

Image:
<image>

Question:
<question>

Here are some candidate answers using different methods.
1. [
Directly answer the question.
<answer1>
]

2. [
First, get the scene graph of the image in JSON format:
<scene_graph>

Then, use the image and scene graph as context to answer the question.
<answer2>
]

3. [
First, the problem can be deconstructed down to sub-questions.
<sub-questions>
<sub-answers>

Then, according to the sub-questions and sub-answers to answer the question.
<answer3>
]

Compare these candidate answers and their solving processes to reflect. Please choose the best candidate answer. You should only answer the number (1, 2 or 3) of candidate answers.

*Answer:*
*<choice>*

Figure 20: Prompts of removing the choice number $n$ in Section 5.5.

Image:
<image>

Question:
<question>

Here are some candidate answers using different methods. They may be right or wrong.
1. [
Directly answer the question.
<answer1>
]

2. [
First, get the scene graph of the image in JSON format:
<scene_graph>

Then, use the image and scene graph as context to answer the question.
<answer2>
]

3. [
First, the problem can be deconstructed down to sub-questions.
<sub-questions>
<sub-answers>

Then, according to the sub-questions and sub-answers to answer the question.
<answer3>
]

According to these candidate answers and their solving processes, generate a more promising answer.

***Answer:***
*<more promising answer>*

Figure 21: Prompts of generating an answer without choosing the best candidate answer in Section 5.5.

