# OpenReview forum: "Self-Choose: Leveraging Diverse Reasoning Solutions to Self-Correct Multimodal Large Language Models"
_ICLR.cc/2025/Conference — ICLR 2025 Conference Withdrawn Submission_

### Official Review · Reviewer_XX94 · 2024-11-01

**Soundness:** 2
**Presentation:** 3
**Contribution:** 2
**Rating:** 3
**Confidence:** 5

**Summary:**

This paper proposes a "self-choose" prompting method that allows the multimodal language model (MLLM) to self-correct during its reasoning process. It is composed of three steps:

1. The model generates three different types of CoT reasoning: direct answering, writing scene graphs for CoT (CCoT), and decompose-then-answer (DDCoT).
2. Then, the model is asked which answer is the best by directly generating the option number (1, 2, or 3)
3. If all answers are deemed wrong, the model is asked to generate a more promising answer.

The method is tested on three multimodal benchmarks: ScienceQA, WHOOPS and MM-Vet. The model outperforms baselines such as using any single CoT method, self-consistency, multi-agent debate, and meta-reasoning prompting.

**Strengths:**

1. The authors showed that previous self-correct methods struggle to improve the reasoning performance on MLLM, a similar finding with Huang et al. [1] who did a similar study in text-only reasoning tasks.
2. The authors modified the previous self-correct approaches (which they called self-refine and self-review) and improved the reasoning performance of MLLM on multiple tasks.


[1] Huang et al. Large language models cannot self-correct reasoning yet.

**Weaknesses:**

1. The contribution is incremental. This paper is basically "using different prompting methods to do CoT and then prompt the model to select the best one". This approach is like a combination of **self-consistency** (sampling multiple answers and find the majority) and **self-review** (asking the model if the generated answer is correct; if wrong then write a new one). All the components in this pipeline have similar ideas in the literature and are very easy to think of based on the prior research works. This pipeline just combined these components and applied it in the multi-modal scenario.\
For example, the idea of "using different CoT prompts to sample answers" can be found in [1,2,3]; the idea of "sampling multiple answers and prompt the LM to select the best" can be found in [4], not to mention it is also an incremental adjustment of the original self-consistency; "writing a more promising answer based on sampled ones" is also a simple adaptation of previous self-correct approaches.

2. The pipeline is not tested on some more representative multi-modal reasoning tasks, to name a few, MMMU (multimodal general reasoning), RAVEN (multimodal logical reasoning), and MathVista (multimodal math reasoning). This weakens the soundness of the experiments. Moreover, the improvement on the tested benchmarks are very limited. It improves less than 1.5 points (on average) compared to only using CCoT without self-consistency (but uses more model calls which hurts efficiency). The performance gap is even smaller compared to some variants in the ablation study (Table 4), which questions the effectiveness of the components in this pipeline. I wonder if some significance test can be performed, e.g., using different random seeds for answer sampling.

3. It looks like the approach is not specifically for multimodal tasks but can also be applied to single-modal (text-only) tasks, whereas the authors only provided a very simple comparison in Appendix E on GSM8k. They only compared to CoT and least-to-most reasoning without other more similar methods like self-consistency. I don't think this is solid enough to show the generalization of this method.

[1] Orca 2: Teaching Small Language Models How to Reason
[2] MathPrompter: Mathematical Reasoning using Large Language Models
[3] MinT: Boosting Generalization in Mathematical Reasoning via Multi-View Fine-Tuning
[4] Universal Self-Consistency for Large Language Model Generation

**Questions:**

N/A

---

### Official Review · Reviewer_rPNM · 2024-11-02

**Soundness:** 2
**Presentation:** 3
**Contribution:** 1
**Rating:** 3
**Confidence:** 4

**Summary:**

This paper presents a self-correction strategy for MLLMs called Self-Choose. The proposed method addresses the inherent reasoning errors in MLLMs by leveraging diverse reasoning methods to generate multiple candidate answers. Through comparison of the reasoning processes and outcomes, Self-Choose selects the most promising answer or generates an improved solution if all initial answers are inaccurate. This approach, tested on benchmarks like ScienceQA, WHOOPS, and MM-Vet with models such as LLaVA and Gemini, shows improvement in reasoning accuracy.

**Strengths:**

- The Self-Choose method provides an effective alternative to existing self-correction strategies like Self-Refine and Self-Review, which is also supported by the experimental results
- The paper is well-written and easy to follow

**Weaknesses:**

- The paper claims to focus on multimodal reasoning, but the research questions investigated do not inherently necessitate a multimodal context. The methods introduced appear to have broader applicability and could potentially be effective in a text-only setting as well. Also, the analysis and case studies in this paper also do not emphasize that why multimodal context is essential.
- Based on point 1, the paper lacks a comprehensive analysis that evaluates the effectiveness of these methods in a non-multimodal context (they include the preliminary results on GSM8K in appendix, which should be placed in the main context). This omission weakens the justification for a multimodal focus. To strengthen the paper, the authors should either conduct more comprehensive experiments to assess their methods' performance in a non-multimodal setting or provide a clearer rationale in the introduction for why multimodal reasoning is essential for this study.
- The benchmarks chosen for evaluation in this paper (ScienceQA, WHOOPS, MM-Vet) do not comprehensively cover multimodal and reasoning-intensive tasks. Widely recognized multimodal math reasoning benchmarks such as Geometry 3K and MathVista should be included to more accurately assess and validate the effectiveness of the proposed methods.
- The proposed self-choose approach offers limited performance improvement over the All CCoT baseline. This marginal gain may not justify its significantly higher inference cost, especially when compared to the baseline's efficiency. Further optimization or a more compelling trade-off between performance and inference cost would strengthen the overall value of the self-choose method.

**Questions:**

- The table caption should be placed above the table content
- Figure 5 is not clear to the readers

---

### Official Review · Reviewer_BNHH · 2024-11-05

**Soundness:** 2
**Presentation:** 3
**Contribution:** 1
**Rating:** 3
**Confidence:** 4

**Summary:**

The paper presents a Self-Choose strategy to teach MLLMs to use diverse reasoning solutions to self-correct. Evaluation shows that self-choose achieves improvements over previous reasoning prompts.

**Strengths:**

The proposed method is simple yet effective, and the experiments show self-choose's great potential.

**Weaknesses:**

Since self-consistency (https://arxiv.org/abs/2203.11171), self-refine (https://arxiv.org/abs/2303.17651) and self-correct (https://arxiv.org/abs/2310.01798) have been proposed in the field of large language models, **extending these strategies into multimodal LLMs and gaining performance gives minor contributions.**

The difference between this method and the cited important baselines (i.e. CCoT (CVPR 2024) and DDCoT (NeurIPS 2023) is that **previous works have tackled the specific challenge of multimodal reasoning**. For example, CCoT uses the scene graph of image, and DDCoT divides tasks between LLMs and MLLMs.

**Unlike these**, the method in this work can not only be applied to multimodal reasoning, but also be directly applied to text-based reasoning. Sampling multiple reasoning paths and aggregating them **have been explored** in text-based reasoning scenarios  (e.g. ReConcile (https://arxiv.org/abs/2309.13007), Peer Review (https://arxiv.org/abs/2311.08152, https://arxiv.org/abs/2307.02762 )).

Therefore, I think the contribution of this work is limited.

**Questions:**

- What is the robustness of this method? For the reported results in Table 3, how many times  was the experiment run?
- What is the version of Gemini-Vision? When were the experiments conducted?
- What are the inference costs of Self-choose compared to IO-SC, CCoT-SC, DDCoT-SC?
- Table 1 shows that the performance of self-refine and self-review degrades when increasing the number of rounds. What about your proposed self-choose method?

---

### Official Review · Reviewer_MURc · 2024-11-09

**Soundness:** 2
**Presentation:** 3
**Contribution:** 2
**Rating:** 3
**Confidence:** 4

**Summary:**

The authors propose a prompting strategy called Self-Choose to teach models to self-correct without any training or external feedback. This strategy explicitly prompts the models to answer with different reasoning paths and generate a new answer if all candidate answers from the reasoning paths are deemed inaccurate. The authors evaluate their methods on Gemini-vision and LLaVA-1.6-13b and conduct comprehensive ablation studies.

**Strengths:**

1. The paper develops a simple plug-and-play self-correct prompting strategy named Self-Choose that does not require training or fine-tuning.
2. The authors present interesting conjectures, making connections with psychology, of why they think Self-Refine and Self-Review do not work.
3. The authors provide structured formulations of their proposed method.
4. The authors conduct comprehensive ablation studies with various variables within Self-Choose.

**Weaknesses:**

1. My primary concern is novelty, especially in the popular field of prompting strategies for MLLMs.

    (a.) Firstly, this paper presents limited literature reviews and comparison studies in the self-correct domain to effectively distinguish itself from existing methods.

    (b.) Secondly, the idea of having multiple reasoning paths is not new and resembles ToT[1], SCoT[2], StrategyLLM[3], etc., which the authors do not address at all in the paper.

    (c.) Finally, this method is overly simplified and straight-forward. With existing methods that utilize the search engine (Self-Ask[4]), APIs (Self-Correct[5]), and Python interpreters (Self-Debug[6]) that produce more significant experiment results, this work presents more of an engineering solution than a research advancement.

2. The baselines are very limited. The paper only compares Self-Choose against Self-Refine and Self-Review, where Self-Review is an intermediate improvement that the authors come up with on top of Self-Refine.

3. The results are insignificant. For example, Self-Choose improves LLaVA-1.6-13b on ScienceQA, of 2017 QAs, by 68.86%-67.63%=1.23%. Especially when the model temperature is set to 0.3 (mentioned in Appendix C), I remain skeptical of the significance of the improvements and thus the effectiveness of such a method.

4. Number of models tested are limited. Only Gemini-vision and LLaVA-1.6-13b are tested with these methods. It is unclear if the improvements can be generalized to other MLLMs of various sizes.

5. The claim in L62-63 “the model fails to correct itself with a fixed thinking pattern” is also derived from an analogy to psychological phenomenon and has no scientific studies to back it up.

    (a.) In Table 4, the ablation study “w/o processes s_i” presents very similar performance to Self-Choose: 68.81% vs. 68.86% on ScienceQA, 62.00% vs. 62.65% on WHOOPS. If no thinking processes are presented to the MLLMs and they still achieve similar results, it means that it is not the thinking processes that are boosting the performance, which contradicts with the authors claim that they have “fixed thinking pattern”.

6. It is unclear why MAD and MRP are introduced in section 5.4.3 and how their results affect authors findings.

7. In L104-106, authors mention that they “conduct experiments applying self-correction techniques originally designed for LLMs to MLLMs”. It is unclear how this step is done.

8. It is unclear how LLMs are utilized as an evaluator for experiments on WHOOPS and MM-Vet as mentioned in L291-292.

9. In the ablation study “Generate”, the authors prompt the models saying that “all candidate answers are inaccurate” which is misleading to the models and not an effective ablation study, as the setup falls back on the issues with Self-Refine. A better ablation would be asking the model to generate at all times regardless of the candidates’ correctness.

[1] Yao, Shunyu, et al. "Tree of thoughts: Deliberate problem solving with large language models." Advances in Neural Information Processing Systems 36 (2024).\
[2] Wang, Yu, et al. "Strategic Chain-of-Thought: Guiding Accurate Reasoning in LLMs through Strategy Elicitation." arXiv preprint arXiv:2409.03271 (2024).\
[3] Gao, Chang, et al. "Strategyllm: Large language models as strategy generators, executors, optimizers, and evaluators for problem solving." arXiv preprint arXiv:2311.08803 (2023).\
[4] Press, Ofir, et al. "Measuring and narrowing the compositionality gap in language models." arXiv preprint arXiv:2210.03350 (2022).\
[5] Welleck, Sean, et al. "Generating sequences by learning to self-correct." arXiv preprint arXiv:2211.00053 (2022).\
[6] Chen, Xinyun, et al. "Teaching large language models to self-debug." arXiv preprint arXiv:2304.05128 (2023).

**Questions:**

1. In L49-50 where authors cite Kim et al., 2023, what is the relationship between this work and self-refine? This work is never brought up again in the later sections either.
2. In L49-50 where authors claim that “it fails to correct vision reasoning”, how do the authors reach this conclusion? And how does it fail?
3. In Appendix D, the authors mention that Self-Choose with LLaVA-1.6-13b on ScienceQA incorrectly changed 1.54% original answers from right to wrong and 2.73% from wrong to right. This should mean that the overall performance increase is 2.73%-1.54%=1.19%, but why is it 68.86%-67.63%=1.23% in Table 3? What are the original numbers of correct and wrong answers from the experiments?
4. Is there a particular reason that the authors mention MAD and MRP as comparisons? And what are the findings?
5. How do the authors “conduct experiments applying self-correction techniques originally designed for LLMs to MLLMs” described in L104-106? Any new visual prompting methods?
6. How do the authors utilize LLMs as evaluators for experiments on WHOOPS and MM-Vet? While LLMs as evaluators remain a controversial approach, are there any remedies taken to prove its effectiveness in authors use cases such as Cohen’s Kappa?
7. How are the reasoning methods selected? It seems rather arbitrary in the paper.
8. What is the “self-reflection” issue referring to in L404?
9. How do the models determine what are “wrong candidates” in the (1) and (2) of “Other settings for Equation 4” without human feedback?
10. What is the “Self-Debate” referring to in L516-517? This is the only place that this word has come up.

---

### Note · Authors · 2024-12-17

I have read and agree with the venue's withdrawal policy on behalf of myself and my co-authors.